Mapping desert shrub aboveground biomass in the Junggar Basin, Xinjiang, China using quantile regression forest (QRF)

Yang XueFeng 107622001010016@xjnu.edu.cn 744157426@qq.com
1 Xinjiang Laboratory of Lake Environment and Resources in Arid Zone , Urumqi , Xinjiang Uygur Autonomous Region , China
2 College of Geographic Science and Tourism, Xinjiang Normal University , Urumqi , Xinjiang Uygur Autonomous Region , China
Phairuang Worradorn
Electronic publication date: 2025 Mar 7
Publication date: 2025
Volume: 13
Electronic Location ID: e19099
Received 2024 Nov 27; Accepted 2025 Feb 12
Copyright: ©2025 Yang
Copyright year: 2025
Copyright holder: Yang
License: This is an open access article distributed under the terms of the Creative Commons Attribution License, which permits unrestricted use, distribution, reproduction and adaptation in any medium and for any purpose provided that it is properly attributed. For attribution, the original author(s), title, publication source (PeerJ) and either DOI or URL of the article must be cited.
License URL: https://creativecommons.org/licenses/by/4.0/

Keywords: Aboveground biomass, Haloxylon ammodendron, Remote sensing, Junggar basin, Uncertainty map

Funding: Natural Science Foundation of Xinjiang Uygur Autonomous Region, China 2022D01A97 National Natural Science Foundation of China 42261062 This research was supported by the Natural Science Foundation of Xinjiang Uygur Autonomous Region, China (Grant No. 2022D01A97) and the National Natural Science Foundation of China (Grant No. 42261062). The funders had no role in study design, data collection and analysis, decision to publish, or preparation of the manuscript.

==============================
Haloxylon ammodendron is an essential species within the Central Asian desert ecosystem, with its aboveground biomass (AGB) serving as a crucial marker of ecosystem health and desertification levels. Precise and effective methods for predicting AGB are vital for understanding the spatial distributions and ecological roles of desert regions. However, the low vegetation cover in these areas poses significant challenges for satellite-based research. In this study, aboveground biomass training and validation datasets were gathered using UAV LiDAR, covering an area of 50 square kilometers. These datasets were integrated with high-resolution, multi-temporal satellite images from Sentinel-1 (S1) and Sentinel-2 (S2). This study applied a spatial cross-validation method to develop a quantile regression forest (QRF) prediction model. This model was used to predict the AGB of Haloxylon ammodendron forest across a study area of 14,000 square kilometers. The findings suggest that, when supported by ground data, multi-source remote sensing technology can estimate the AGB distribution of Haloxylon ammodendron over large areas. Significant uncertainty exists within the model due to the low vegetation cover characteristic of arid regions and the uneven distribution of sampling points. This uncertainty can be reduced by using area of applicability (AOA) and uncertainty maps, which identify the regions where the model’s predictions are most accurate and guide further data collection to enhance precision. This study provides improved insight into the spatial distribution and extent of Haloxylon ammodendron AGB in the research area and offers essential geospatial information for ecosystem conservation strategies. The results also contribute to the understanding of how desert vegetation growth and carbon cycling respond to environmental changes, and for forecasting future vegetation dynamics in arid regions.

Introduction

Haloxylon ammodendron is known for being a significant species in the afforestation process and a valuable C4 woody shrub in arid regions. It exhibits remarkable resilience to harsh environmental conditions like high temperatures, drought, and salinity. It is widely distributed in central Asia, occupying 10% of the arid land in northwest China (Guo et al., 2005). Haloxylon ammodendron ranges in size from a large shrub to a small tree and is often called “Desert Forest” due to its important ecological value. It constitutes approximately two-thirds of the total area of such deserts in China within the Junggar Basin. Here, Haloxylon ammodendron and Haloxylon persicum form sparse single-dominant species communities in typical desert habitats, creating a distinct desert ecological landscape and serving an important role in desert ecosystems (Hu, 1984; Li et al., 2019; Yu et al., 2023).

Terrestrial ecosystems across the globe, particularly those in arid and semi-arid desert regions, are extremely fragile and vulnerable to disruptions, which makes them highly sensitive to climate changes and human activities (Dai, 2011). An essential quantitative metric of desert shrub communities in arid and semi-arid ecosystems is the aboveground biomass (AGB), which is essential for assessing ecosystem productivity and monitoring desertification. The AGB can depict the carbon sequestration status of the local ecosystem and can be used as a quantitative indicator of natural activity and human impacts (Tao & Zhang, 2013). Additionally, estimating the organic carbon pool and studying its dynamics in arid and semi-arid desert ecosystems hold significant scientific importance for regional and global-scale carbon cycling research (Schimel, 2010).

The above-ground biomass of shrubs can be obtained through direct measurements (typically via the harvest method, which requires destructive sampling, drying, and weighing of plant tissues, despite its high labor costs) or indirectly calculated by establishing empirical allometric growth equations between measured aboveground biomass and structural parameters such as diameter at breast height, tree height, or crown width (Buras et al., 2012; Niu, Song & Zhou, 2014). However, it is difficult to obtain a true picture of ecosystem-level conditions on a large scale solely through point surveys.

In the past decade, unmanned aerial vehicles (UAVs) have garnered increased attention in the field of ecology. Utilizing ultra-high-resolution cameras, UAVs can capture optical images at the centimeter level and collect high-density LiDAR point cloud data. This technology offers a valuable approach for mapping shrub AGB on a compact spatial scale (Xie et al., 2023; Zhao et al., 2021; Ding et al., 2022; Mao et al., 2021). The use of UAV technology offers a valuable link between on-site surveys and the observation of the earth via satellites (Mao et al., 2022; Chen et al., 2024).

Satellite remote sensing technology has been widely applied in forest and grassland biomass studies due to its large-scale and high-efficiency characteristics (Lourenço, 2021; Lu et al., 2014). However, the study of shrub biomass has its particularities; shrubs are smaller in individual size and exhibit significant discontinuity and heterogeneity in horizontal distribution with less distinct spectral features (Zandler, Brenning & Samimi, 2015; Wang et al., 2015). Within dry and semi-dry environments, where sparse woody vegetation predominates against a bright soil background, the vegetation canopy spectra contain considerable information about soil (Li et al., 2018), presenting substantial challenges to shrub biomass research. Recent studies have used medium-to-low-resolution satellites such as Landsat, Sentinel-1, Sentinel-2, and MODIS to predict the AGB of Haloxylon ammodendron and desert shrub communities (Chen et al., 2018; Sabaghzade et al., 2017; Batsaikhan et al., 2020; Yan, Wu & Wang, 2015; Chang & Shoshany, 2016). A common method used in this research involves the creation of AGB regression models including the correlation between aboveground biomass and reflectance values in spectral bands or optical vegetation indices. While most studies claim relatively ideal results, there are also instances of failure (Sepehry & Hassanzadeh, 2004), indicating a certain degree of uncertainty in large-scale shrub aboveground biomass monitoring using medium-to-low-resolution satellite data. Although high-resolution satellite imagery can mitigate some of these issues to a certain extent (Adhikari et al., 2017), its acquisition is relatively more difficult. An issue frequently observed in these studies is the absence of compelling methods for validating accuracy. This implies a lesser focus on the uncertainty in AGB prediction, which is a fundamental aspect of spatial modeling that provides a thorough assessment of predictive accuracy. Moreover, little attention is given to the techniques used in determining model parameters or the risk of overfitting AGB regression models.

Ultimately, the accuracy and efficiency of remote sensing methods are crucial for analyzing the spatial arrangement and ecological roles of Haloxylon ammodendron vegetation in arid regions. Nonetheless, research focusing on estimation methods for the AGB of large-scale desert shrubs using remote sensing techniques is relatively scarce. The goal of the current study was to address these gaps in the literature.

Materials and Methods

Research area

This investigation focused on the western sector of the Junggar Basin in Xinjiang, China (Fig. 1A), specifically the ancient lake basin sediment area formed after the contraction of Manas Lake (Fig. 1B). This area hosts the largest Haloxylon ammodendron forest in Xinjiang (Fig. 1C). This region receives less than 80 mm of annual precipitation, with an annual evaporation rate exceeding 3,600 mm. The average temperature in July exceeds 27 °C, with absolute maximum temperatures surpassing 44 °C, while the average temperature in January falls below −19 °C, with absolute minimum temperatures dropping below −43 °C. The average elevation is around 300 m  (Yin, 1993). The terrain is relatively flat with a few small sand dunes less than 10 m high. The area’s vegetation is mainly composed of pure Haloxylon ammodendron forests (Fig. 1D). Over the past half-century, human activities and climate change have caused significant damage to the “desert forest”, which has yet to fully recover. The study area is characterized by high salinity soils, sparse Haloxylon ammodendron stands with an average coverage of 11.58%, and very few seedlings, indicating a declining population. In such an arid environment, the overall low coverage of Haloxylon vegetation and the high proportion of dead plants within the community (Liu et al., 2010; Si et al., 2011) present conservation challenges. To protect this Haloxylon ammodendron forest, a national public welfare forest reserve was established in 2006. However, there is a lack of corresponding biomass survey data in the national forest resource inventories of China. Studying the biomass of the Haloxylon ammodendron forest in this area is crucial for its conservation and for understanding the trends in natural environmental changes as well as the impact of human activities in this region.

Figure 1 Geographical positioning of the research site and the distribution of UAV sampling zones.

(A) The geographic location of the study region and UAV plot. (B) Xinjiang Uygur Autonomous Region, where the study is focused. (C) The canopy height model (CHM) of a plot. (D1–D9) Landscape of the study area.

Research data

Data obtained through satellite-based remote sensing

Data from the Sentinel-1 and Sentinel-2 satellites spanning the period between 2019 and 2023 were used for this study. The European Space Agency (ESA), through the Copernicus Sentinel program, offers Earth Observation (EO) data at no cost from various satellites with diverse sources and wavelengths that encompass both optical and radar capabilities. Introduced in 2015, the Sentinel-2 satellite constellation, specializing in optical imagery, furnishes complimentary remote sensing data at an enhanced spatial resolution of 10 m, which is a marked enhancement compared to the 30-meter capacity of Landsat. Similarly, the Sentinel-1 project, which started in 2014, uses C-band synthetic aperture radar (SAR) for acquiring radar pictures with a multi-look product resolution of around 20 m. Moreover, the Copernicus Sentinel initiative is characterized by a brief revisit interval ranging from 5 to 12 days, during which advancements in satellite technology and techniques contribute to the enhancement of spatial detail in remote sensing data.

For this study, a dataset containing satellite images from Sentinel-1 and Sentinel-2 was compiled using the Google Earth Engine (GEE) platform. This platform provides powerful tools for examining and handling remote sensing data, offering readily-available access to a wide array of remote sensing datasets from sources including Landsat, Sentinel, and digital elevation models (DEM) (Wang et al., 2024; Yan et al., 2023). Clear filtering was conducted on all Sentinel-2 surface reflectance image collections with a maximum cloud cover of 20% from 2019 to 2023 in GEE. It has been noted in previous research that, as there is a scarcity of herbaceous vegetation, the dry season typically presents more pronounced differences in spectral bands within the visible and near-infrared ranges, showcasing distinct contrasts between soil and woody vegetation. These data can be effectively used for the mapping of woody vegetation in arid and semi-arid environments (Wingate et al., 2019; Ludwig et al., 2019).

In addition to the bands provided by Sentinel-2 at resolutions of 10 m and 20 m, calculations were conducted for vegetation indices related to photosynthetic vegetation (PV) and yellowness vegetation indices sensitive to non-photosynthetic vegetation (NPV) (Lei & Wenz, 2009). Included in the greenness vegetation indices are the normalized difference vegetation index (NDVI), modified soil adjusted vegetation index (MSAVI) (Qi et al., 1994), enhanced vegetation index (EVI) (Huete et al., 1997), and shortwave infrared (SWIR) modification to simple ratio (SR) vegetation index, termed the reduced simple ratio (RSR) (Brown et al., 2000). The yellowness vegetation indices include the normalized difference tillage index (NDTI) (Deventer et al., 1997). Compared to using single indices, combining greenness and yellowness indices provides a more comprehensive representation of the compositional impact of Haloxylon vegetation in the study area. Some studies have found that NDTI is suitable for identifying dry vegetation in arid and semi-arid environments (Wu et al., 2021; Rossi et al., 2021), and MSAVI is advantageous in specific application scenarios due to its ability to reduce soil background influence (Qi et al., 1994). By incorporating shortwave infrared into the SR, RSR effectively mitigates the impact of soil background reflectance (Zhu et al., 2010).

Texture metrics of the gray-level co-occurrence matrix (GLCM) were also calculated using the grayscale images derived from the near-infrared (NIR) band (B8), red band (B4), and green band (B3) of Sentinel-2. Texture metrics obtained using GLCM have been recognized as reliable indicators of vegetation structure in previous research (Wood et al., 2012), proving advantageous for remote sensing studies focused on vegetation (Dobrowski et al., 2008; Wang, Lv & Li, 2017).

From the Sentinel-1 ground range detected (GRD) data collection, images were selected from the vertical–vertical (VV) and vertical–horizontal (VH) polarization channels (backscatter coefficient). These images were then cropped and masked according to the area of interest (AOI). For the filtered images, further processing was then conducted based on the condition ‘DESCENDING’. Additionally, slope data was obtained from the Shuttle Radar Topography Mission (SRTM).

This study used seasonal information to select images from each of the following periods: spring, summer, autumn, and winter. For every season, median composite images were calculated to mitigate the effects of outliers or noise.

To summarize, this study selected a total of 10 features of visual, red edge, and SWIR bands; seven GLCM features; five vegetation index features from Sentinel-2; two radar features from Sentinel-1; and two topographic features as independent variables for AGB modeling. Moreover, with the exception of the two topographic features, corresponding values for spring, summer, autumn, and winter were extracted and calculated for all of the other features, resulting in a total of 98 features (Table 1). This presented a representative view of the surface conditions during that period. Subsequently, the resolution of all data was standardized to 10 m. These data constitute the “Original Dataset” of the study area.

Table 1 List of variables for AGB model.

Variable types	Remote sensing variables	Explanation or formulation	Seasonal processing	
Radar features	VV	Vertical-Vertical polarization backscattering coefficient		
VH	Vertical-Horizontal polarization backscattering coefficient	Yes	
Visual, Red edge, SWIR band features	B2	490 nm band of S2	10-meter spatial resolution		
	B3	560 nm band of S2			
	B4	665 nm band of S2			
	B8	842 nm band of S2	Yes	
	B5	705 nm band of S2	20-meter spatial resolution		
	B6	740 nm band of S2			
	B7	783 nm band of S2			
	B8A	865 nm band of S2			
	B11	1,610 nm band of S2			
B12	2,190 nm band of S2		
GLCM features	Cont	Contrast metric	5*5 kernel size		
	Asm	Angular Second Moment			
	Corr	Correlation metric			
	Ent	Entropy metric			
	Var	Variance metric		Yes	
	Idm	Inverse Difference Moment metric			
savg	Sum Average metric		
Vegetation index features	NDTI	Normalized Difference Tillage Index	SWIR 1−SWIR 2SWIR 1+SWIR 2		
	RSR	Reduced Simple Ratio index	NirRed∗1−SWIR−SWIR minSWIR max−SWIR min		
	EVI	Enhanced Vegetation Index	2.5∗Nir−RedNir+6∗Red−7.5∗Blue+1	Yes	
	MSAVI	Modified Soil Adjusted Vegetation Index	2∗Nir+1−2∗Nir+12−8∗Nir−Red2		
NDVI	Normalized Difference Vegetation Index	Nir−RedNir+Red		
Terrain features	Elevation	NASA SRTM Digital Elevation 90 m data		
Slope	Terrain slope from NASA SRTM	No	

Field data

Fifty sampling sites were set up in the study area due to the limitations of traffic conditions. They spanned east–west and north–south directions and covered the period from May to September between 2021 and 2023. Encompassing various levels of degradation, the plots included Haloxylon ammodendron and Haloxylon persicum communities found in sandy plains, stabilized dunes, and semi-stabilized dunes, covering an area of approximately 50 square kilometers.

Drone mapping was conducted using the DJI Matrice 300 RTK with Zenmuse L1 LiDAR surveying capabilities. Operating in ground-based real-time kinematic (RTK) mode, the drone operated at a height of 100 m and traveled at a velocity of eight meters per second. The scanning mode was dual return with repeated scanning, and the laser side overlap rate was 50%. The flight path was automatically executed using the DJI Pilot planning software. The obtained point cloud density was higher than 100 points per square meter.

The data collected underwent processing using the DJI Terra software (DJI Terra, 2024) for the generation of LAS point cloud data. In the ENVI LiDAR module, the point clouds were filtered at a 0.5-meter resolution for the creation of the digital surface model (DSM) and digital elevation model (DEM). The canopy height model (CHM) was then calculated by differencing the DSM and DEM. Finally, CHM segmentation was performed using Lidar360 software (GreenValley International, 2024). To avoid the influence of other small shrubs, all seed points below 0.5 m in height were filtered out. The final outputs included the positions of all individual trees, tree heights, crown area, and vector files of individual tree boundaries.

Methods

This research used data from UAV-LiDAR field surveys, as well as Sentinel-1 and Sentinel-2 imagery, which included SAR backscatter coefficients, texture characteristics, spectral bands, and vegetation index attributes. Employing the quantile regression forest (QRF) prediction method, the Haloxylon ammodendron AGB in the western Junggar Basin was estimated to assess the model’s reliability and the level of uncertainty in the results. The workflow for the study is depicted in Fig. 2 and is expanded upon in the following sections.

Figure 2 Flow chart of estimating Haloxylon ammodendron biomass.

Aboveground biomass calculation

This study focused on Haloxylon ammodendron, the dominant species of the Junggar desert ecosystem. Previous research on the allometric equations for estimating the biomass of Haloxylon ammodendron tree often used power function models with parameters such as basal diameter, crown area, and tree height (Hemmati, Kiani & Arani, 2018; Song & Hu, 2011). Considering that UAV LiDAR cannot directly obtain the basal diameter, the estimation model for calculating the AGB of individual Haloxylon ammodendron trees was chosen to include only crown area and tree height as indicators (Eq. (1); where C is the canopy area and H is the tree height) (Tao & Zhang, 2013). (1) AGB=0.3628∗CH0.9605.

Satellite data were analyzed using the “Zonal Statistics as Table” function with hexagonal grid cells as units, calculating the average value of the variables for each cell. The Haloxylon ammodendron forest parameters (cover, density, and height) for each cell were obtained by performing a spatial overlay operation between the Haloxylon ammodendron tree vector files and the hexagonal grid cells. Biomass was calculated and summed using Eq. (1). The entire process was implemented using ArcPy.

Resampling strategy

This study used hexagonal grids as the units for AGB statistics and modeling. Hexagons are the shapes closest to circles that can tessellate a plane regularly. Compared to squares, they possess additional symmetry, which can reduce sampling bias caused by edge effects of grid shapes (Birch, Oom & Beecham, 2007; Birch, Vuichard & Werkman, 2000). Hexagonal grids have also been used in AGB survey research (Bruening et al., 2023).

The determination of grid cell size was influenced by several factors, the most significant being a noticeable misalignment in the registration of data between Sentinel-1 and Sentinel-2 images (Ye et al., 2021); Secondly, there are spatial coordinate registration errors between the UAV ground sampling area and images from Sentinel-1 and Sentinel-2. Expanding the spatial coverage of sample units can mitigate the effects of common registration errors on model precision (Hogland & Affleck, 2019). This refinement in granularity can also minimize variations arising from spatial mismatches between the sampling data and Sentinel data at the sub-pixel level (Shafeian, Fassnacht & Latifi, 2021). As the sampling grid radius expands, larger plots help maintain a heightened level of spatial correspondence between the ground reference datasets and the the remote sensing datasets, potentially improving the AGB model’s performance (Frazer et al., 2011). In this investigation, hexagonal grid cell sizes were tested from 100 square meters to 800 square meters. AGB estimates obtained through random forest prediction were compared to reference AGB using R2 analysis. Figure 3 presents violin plots that depict the model performance metrics obtained during the iterative evaluation of the RF model for AGB estimation. The findings reveal that R2 values vary from the highest resolution of 300 square meters to 800 square meters, with the highest mean R2 at 600 square meters. However, the R2 values then slowly stabilizes and remains roughly constant. Considering that finer spatial granularity is deemed beneficial, a grid cell size of 600 square meters was used for modeling.

Figure 3 Grid scale analysis.

Cross-validation

Cross-validation (CV) techniques can be used to effectively assess model performance. Spatial data often demonstrate spatial autocorrelation, in which nearby observations can have an impact on each other. Relying solely on random CV could result in inaccurately low error estimates. However, random CV strategies fail to assess the model’s effectiveness in spatial mapping, and the validation process is limited to the evaluation of its capability to replicate the gathered information  (Meyer et al., 2019; Ploton et al., 2020). Taking into consideration the spatial structure in both the training and test sets, spatial CV methods aim to ensure similarity, thus enhancing the model’s accuracy in generalization assessment.

This research used a K-fold spatial CV, which divided the spatial data into K separate subsets, each preserving the spatial structure integrity. This process is commonly accomplished through techniques including spatial stratified sampling or spatial K-means clustering. Following this approach, each subset was used consecutively as the testing set, while the remaining K-1 subsets were designated for training. This process involved conducting multiple rounds of model training and evaluation, which culminated in the aggregation of final evaluation results. The implementation involved iterative training of the model through random 10-fold CV, followed by the assessment of its predictive performance using the complementary fold data (Linnenbrink et al., 2024; Milà et al., 2022; Valavi et al., 2019). Spatial CV was performed by using the K-fold nearest neighbor distance matching (kNNDM) function from the R CAST package.

Feature selection

The careful selection of features is critical in machine learning and data analysis. This process involves selecting the most useful features (or variables) from the raw data for the modeling process, with the aim of improving model performance, reducing overfitting, enhancing model interpretability, and decreasing training time (Guyon & Elisseeff, 2003). The study employed the Boruta method for selecting features. This algorithm generates randomized duplicates of each input variable, compares these shadow features with the original features, and uses an evaluation metric (e.g., Z-score) to determine which features significantly outperform the shadow features. These features are considered important and are deemed relevant to the target variable (Kursa & Rudnicki, 2010). Implementation of the R Boruta package was used in executing the Boruta algorithm.

AGB modeling

Random forest (RF), an ensemble learning method composed of multiple decision trees, is widely used for AGB estimation (Chen et al., 2018; Tamiminia et al., 2021). Random subsets from the training data are used to enhance the model’s diversity. The unselected data, known as “out-of-bag” data, are used for internal error validation, ensuring the model’s robustness across different datasets. At each split, the most effective predictors are randomly chosen to decrease the correlation among the regression trees. Each tree is grown to its fullest extent, and the final prediction is made by averaging the outputs from all the trees (Breiman, 1996).

Quantile regression forest (QRF) is an extension of random forest that retains all prediction values for each observation. This enables QRF to assess the conditional distribution and estimate various quantiles. By examining the span between the highest and lowest quantiles for an individual forecast, it is possible to depict the uncertainty associated with that forecast. The application of QRF has been demonstrated in numerous research papers (Meinshausen, 2006; Athey & Imbens, 2016; Khanal et al., 2023). In the current study, the 0.5 quantile was chosen to represent the AGB prediction value, while the 0.05 and 0.95 quantiles were selected to depict the range of uncertainty levels, which aided in a thorough assessment of the prediction outcomes’ confidence intervals. This study employed the R caret package to implement QRF.

Area of applicability

The rise in popularity of machine learning algorithms in environmental spatial mapping stems from the capacity these algorithms have for accommodating intricate and nonlinear relationships. However, the applicability of these algorithms is limited to data that closely resembles the training dataset. When machine learning algorithms make predictions for areas with insufficient training data, the results of these algorithms frequently deviate significantly from the expected margin of error. Furthermore, it is common for new geospatial regions to introduce novel predictive characteristics. As a method to assess the applicability of prediction models in different regions, the area of applicability (AOA) approach was used in this research (Meyer & Pebesma, 2021; Gazis & Greinert, 2021).

The AOA is attained by calculating the dissimilarity index (DI) relative to the nearest training data points within the multi-dimensional predictor variable space. The importance of each predictor variable is considered in the model’s weight assignment process. Subsequently, the DI undergoes a thresholding process to determine the AOA (Ji et al., 2024). In this study, the DI calculation and AOA derivation were conducted using the R CAST package.

Results

Resample result

Using ArcPy, approximately 120,000 hexagonal grid cells were continuous generated within the boundary of the UAV sampling area, each with an area of 600 square meters. The height, density, and coverage data of Haloxylon ammodendron in each unit was calculated, as well as the biomass of Haloxylon ammodendron in each unit according to the allometric growth equation (Eq. (1)). Next, from the “Original Dataset”, the median values of 98 features corresponding to each hexagonal grid cell were extracted to reduce the impact of outliers or noise. The “AGB DataSet” is constructed using the following data from all grids: the biomass of Haloxylon ammodendron, the 98-features data, and the center point coordinates (xCentroid and yCentroid).

The statistical grid aggregates data from all units to obtain the distribution percentages for cover (Fig. 4A), density (Fig. 4B), height (Fig. 4C), and biomass (Fig. 4D).

Figure 4 Field data analysis.

(A) Cover. (B) Density. (C) Height. (D) Biomass of field data.

Statistical analysis revealed that the indicator values for most sampling units were generally low. For example, the percentage of units with less than 10% cover was 94.47%, the percentage with a density fewer than 20 plants was 93.22%, the percentage with an average height less than two m was 94.16%, and the percentage with an AGB less than 50 kg was 95.76%. This indicated that the basic nature of the ecosystem distribution of Haloxylon ammodendron in the area under examination had an overall low plant density, predominantly sparse vegetation, and only isolated areas of high density.

CV results

Fifty percent of the data was randomly sampled from the “AGB DataSet”, and the createFolds function from the R CAST package was then used to generate a 10-fold random CV dataset (Fig. 5A). These sampling points were relatively scattered, without any obvious spatial clustering patterns or regularities. For comparison, the kNNDM function from the R CAST package used the xCentroid and yCentroid of the “AGB DataSet” to create a 10-fold spatial cross-validation dataset (Fig. 5B). In comparison with the results of random sampling, the distribution of these sampling points exhibited a certain spatial structure or pattern.

Figure 5 Comparison of random cross-validation and spatial cross-validation.

(A) Ten-fold random cross-validation dataset. (B) Ten-fold spatial cross-validation dataset. (C) Distance density plot of random CV. (D) Distance density plot of Spatial CV.

The GEODIST function was used to calculate the nearest neighbor distances between prediction locations and training locations under different sampling scenarios to further assess the reliability of predictions. The calculations performed by the GEODIST function included the assessment of the nearest neighbor distances between prediction and training points (prediction-to-sample), the distances between test and training points in a leave-one-out cross-validation (LOO CV) scenario (sample-to-sample), and the distances between test and training points in a specific CV setup (CV-distances). In an ideal scenario, the distances observed throughout CV would closely mirror those seen in the prediction phase for the predictive conditions to be accurately reflected in the cross-validation process.

Under random sampling conditions, the distance between training points in terms of geographical nearest neighbor distance (NND) is significantly shorter than the distance between prediction points and training points (Fig. 5C). In this instance, due to the aggregation of sample locations, the distance between predictions and samples becomes significantly greater than that observed in cross-validation, resulting in notable overfitting of the model. This results in strong training performance with random sampling, but poor performance when predicting in unknown areas (Domingos, 2012). The model’s predictive capability is limited by random CV within the boundaries of the training data. Although the geographic distance distributions between test points and training points and between prediction points and training points do not perfectly match under spatial CV (Fig. 5D), there is a considerable overlap between them. This indicated that the prediction results from spatial CV were closer to real-world values.

Results of feature selection

Fifty percent of the data was randomly sampled from the “AGB DataSet”, and the kNNDM function was used to perform a 10-fold spatial cross-operation to obtain 10 datasets. Next, nine subsets of each dataset were selected as the training set, and the remaining one subset was used as the validation set. The Boruta algorithm was applied to evaluate the feature importance of the data, with the number of iterations set to 150 and the confidence level set to 0.01. The importance scores of different features in multiple iterations were calculated using the median. The feature importance scores of multiple folds were integrated and averaged, and the feature importance was sorted. Significant features were identified based on their mean significance levels. With the exception of seven GLCM indices identified in winter, all other 91 features were associated with AGB. Among the various vegetation indices, the NDTI index was identified as the most crucial, succeeded by RSR, EVI, MSAVI, and NDVI. The importance of spectral bands ranked from highest to lowest was B12, B11, B2, and B3 reflectance (Fig. 6A). Elevation also significantly influenced the AGB of Haloxylon ammodendron forest in the study area, with VH having a greater effect than VV. Among the GLCM texture factors, contrast, inverse difference moment, and sum of average were notably important. When accumulating the importance of seasonal factors, autumn factors had the highest cumulative importance, and although winter factors were fewer in number, they ranked high in importance (Fig. 6B).

Figure 6 Analysis of the feature selection process.

(A) Top 30 features in descending order of importance as determined by the Boruta algorithm (“B” represents Sentinel-2 bands, “contr” represents the GLCM Contrast, “idm” represents the Inverse Difference Moment, and “savg” represents the Sum of Averages). (B) Total importance of all 30 features with seasons (0, 1, 2, and 3 correspond to the seasons of spring, summer, autumn, and winter, respectively). (C) Correlation heatmap of the final 12 features.

To further enhance the interpretability of the model while reducing its size and computational load, the top 30 variables from the Boruta importance scores were selected. Highly correlated variables (r >  ± 0.75) were identified, and the less important factors were excluded, resulting in 12 predictor variables (Fig. 6C). These 12 selected predictive variables and the corresponding biomass data were combined into the “Predictor DataSet”.

Results of DI and AOA

Continuous hexagonal grids of 600 square meters were generated for the entire research area and 12 features of each grid from the “Original Dataset” were calculated to constitute the “Predicted Dataset”. Using the AOA function in the CAST package, a dissimilarity index (DI) was calculated based on distances to the training data in the multidimensional predictor variable space. If a certain predictive data point had similar attributes to the training data, its distance in the predictor variable space would be relatively low (DI approaching 0), while the DI at locations with a significant difference in attributes would be relatively high. The computation of a threshold value determined the area of applicability (AOA). The dissimilarity measure of a data point was calculated by dividing its minimum distance to the training data by the average distance. The final threshold value was determined by multiplying the IQR by 1.5 and adding this amount to the 75th percentile value. This specific threshold served as a benchmark for determining if new data points aligned with the training data, ultimately establishing whether they fell within the model’s AOA. The binary map generated by the AOA analysis distinguished regions as either within the AOA (designated as ‘1’), where model predictions are reliable, or outside the AOA (designated as ‘0’), where model predictions become uncertain, as illustrated in Fig. 7.

Figure 7 Area of application (AOA).

The AOA analysis indicated that approximately 35% of the study area differed from the training samples. This included regions such as the southern oasis, the downstream wetland of the Manas River (Mayi Lake), the Dabasan Nuur Salt Lake, the Small Salt Lake and its surrounding areas, the northwestern piedmont Gobi, and parts of the eastern Gurbantünggüt Desert. In these regions, land cover significantly deviated from the sampled areas, thus falling outside the model’s applicability. The remaining 65% of the region was suitable for model prediction.

AGB uncertainty map

Fifty percent of the data was randomly sampled from the “Predictor DataSet”, and 10-fold data was generated using both random and spatial CV methods, which served as the training control data for the quantregForest algorithm. The evaluation metric was set as R-squared to evaluate performance. As a result, AGB Models were constructed using both random and spatial CV methods. The criteria chosen for assessment included coefficient of determination (R2; Eqs. (2) and (3)), root mean squared error (RMSE; Eq. (4)), mean absolute error (MAE; Eq. (5)), symmetric mean absolute percentage error (SMAPE; Eq. (6)), and relative absolute error (RAE; Eq. (7)) (Chicco, Warrens & Jurman, 2021; Correndo, 2024). The calculation methods for each metric were as follows: (In these equations, the predicted ith value was denoted as Xi, while the actual ith value was represented by the Yi element).

(2) Y¯=1m∑i=1mYi

(3) R2=1−∑i=1mXi−Yi2 ∑i=1mY¯−Yi2

(4) RMSE=1m∑i=1mXi−Yi2

(5) MAE=1m∑i=1m|Xi−Yi|

(6) SMAPE=100%m∑i=1m|Xi−Yi||Xi|+|Yi|/2

(7) RAE=∑|Xi−Yi|∑|Yi−Y¯|.

The range of positive coefficient of determination (R2) values is from 0–1, where a value of 1 indicates an accurate prediction. SMAPE (%) is standardized, dimensionless, and bounded, with values ranging from 0–2, where 0 represents a perfect prediction and 2 represents the worst possible prediction. This evaluation metric, RAE, involves a calculation that compares the absolute errors of predictions to the absolute deviations of observations from their mean, and a RAE of < 1 indicates that the model’s predictive ability is better than the baseline model. MAE and RMSE are suitable for comparison across different regression models, while R2, SMAPE, and RAE can be used to evaluate regression quality in absolute terms (Chicco, Warrens & Jurman, 2021).

Although the random CV model results appeared ideal based on the metrics (Fig. 8A), this approach resulted in the concealment of overfitting and generated an excessively positive evaluation of predictive capability. In contrast, the spatial CV model focused on anticipating new clusters within the feature space in each round of training, which led to a reduction in spatial dependence between training and testing observations, resulting in the spatial CV model demonstrating significantly higher MAE, RMSE, RAE, and SMAPE, along with a notably lower R2 compared to the random CV model (Fig. 8B). Nonetheless, the spatial CV model approach aids in averting over-extrapolation and offers a more precise evaluation of the model’s dependability in authentic scenarios. Therefore, despite the spatial CV metrics appearing inferior to those of random CV, the results these metrics provide are more valuable for reference.

Figure 8 Comparison of model metrics between two different types of cross-validation methods.

(A) Random cross-validation and (B) Spatial cross-validation, as illustrated by MAE and RMSE values in kg/plot, with a plot size of 600 m2.

The data in the “Predicted Dataset” with an AOA value of 1 were added into the AGB Model, which was developed using the spatial CV method, in order to predict the spatial distribution of Haloxylon ammodendron forest AGB throughout the entire study area (Fig. 9A).The results showed that the biomass of Haloxylon ammodendron forest exhibited significant spatial heterogeneity within the study area, with a trend of increasing from northwest to southeast. Higher biomass concentrations were discovered in the southeastern section of the study area, near the Gurbantünggüt Desert boundary, as well as in the central region, while the western and northern parts showed relatively lower biomass levels. Certain areas in the southern part also exhibited high biomass concentrations. The range of values predicted by the QRF model’s 0.05 and 0.95 quantiles was used to represent the uncertainty of the prediction results (Fig. 9B). The areas surrounding the cultivated land in the southern oasis had the highest uncertainty. Specifically, higher uncertainty is observed in the region located to the southeast of Dabasan Nuur Salt Lake. This significant uncertainty can be attributable to the insufficient training data available for these areas and the potential existence of substantial AGB in these territories. It is imperative to collect more samples in future research for enhanced prediction accuracy.

Figure 9 The final result map.

(A) AGB Map. (B) Uncertainty Map.

Considering the complex heterogeneity of the ecological environment in the study area, and despite some degree of uncertainty in the predictions, the AGB map still effectively reflects the overall spatial pattern and local characteristics of Haloxylon ammodendron forest AGB within the study area.

Discussion

Analysis of variables

This study used all visible, near-infrared, and shortwave infrared bands at a 10–20-meter resolution from Sentinel-2, five derived vegetation indices, seven texture metrics, and the dual-polarization (VH and VV) backscatter measurements from Sentinel-1. Spatial CV used the Boruta algorithm to analyze the significance of each element. A high rank in importance was achieved by bands B11 and B12 in the shortwave infrared (SWIR) spectrum, along with the NDTI and RSR indices, which incorporate SWIR wavelengths. The high importance of these bands and indices may be linked to the distinctive biophysical characteristics of Haloxylon ammodendron trees found in the study area.

Due to the extremely arid climatic conditions of the study area, the leaves of Haloxylon ammodendron trees in this area have degenerated into scales, and their branches have grown as assimilating branches to perform photosynthesis. The chlorophyll area mass gradually decreases with the lignification process, where the mass of lignin and cellulose increases. Moreover, a certain proportion of the community consists of dead, standing plants (NPV). The infrared and near-infrared bands, along with common vegetation indices (e.g., NDVI, MSAVI), cannot effectively distinguish NPV from soil (Li, 2017), thus losing the capacity to estimate AGB. In environments with sparse canopy cover, the presence of bare soil significantly reduces the distinctive spectral features of photosynthetic vegetation (green plants). Additionally, the near-infrared and visible spectral bands display a noteworthy resemblance between soil and non-photosynthetic vegetation (e.g., dead plants and litter), with differences mainly observed in reflectance levels at specific wavelengths (Ren & Zhou, 2019), thus restricting the significance of the near-infrared and red bands.

The 2,000–2,400 nm shortwave infrared bands are sensitive to lignin and cellulose and are characteristic spectra for distinguishing dry vegetation, bare soil, and exposed bedrock (Cao et al., 2020). Studies have shown that the SWIR bands also exhibit some sensitivity to vegetation height and biomass (Cunliffe et al., 2020), while the SWIR spectrum shows sensitivity to the water content of canopy leaves within forest formations (Zhao et al., 2022). The NDTI, calculated using two different shortwave infrared bands (SWIR1 and SWIR2), effectively reduces the interference of soil background on vegetation indices, thereby more accurately reflecting vegetation conditions. This sensitivity makes the NDTI highly effective in distinguishing surface cover (such as crop residues and vegetation cover) from bare ground (Dai et al., 2018).

The role of the reduced simple ratio (RSR) is also significant. Vegetation indices such as EVI and MSAVI, which reduce the impact of soil background, are also prominent. The importance of elevation reflects the primary trend of Haloxylon ammodendron forest AGB increasing from the northwest to the southeast of the study area. This geomorphological transition—from low-lying areas (silty plains) and small dunes (less than five meters in height) to large dunes (greater than 10 m in height)—aligns with previous studies that have observed large-scale decline of Haloxylon ammodendron populations in flat terrain and small dunes, whereas those in larger dunes tend to thrive relatively well (Sun, 2018).

The role and importance of multi-temporal data

Included in the 12 factors ultimately used in the regression model of this study were two spring factors, two summer factors, four autumn factors, three winter factors, and elevation. In previous studies, data from the autumn and winter seasons were not customarily used. The prominent role of seasonal factors in this study might be attributable to the unique floristic composition of ephemeral plants in the Junggar Basin desert. In years with substantial winter and spring precipitation, the overall vegetation cover can exceed 70%, creating a lush green spring “meadow” appearance, which is significantly different from other deserts (Pan & Zhang, 1996). This spring seasonality of sand vegetation is unique to this region. Spring and summer are the growing seasons for these ephemeral plants, during which their vigorous growth reduces the difference between Haloxylon ammodendron trees and the soil background, altering the ecosystem’s structural composition and potentially causing signal confusion (Yuan & Tang, 2010; Chen, Zhang & Hu, 1983). In autumn and winter, ephemeral plants in the Junggar Basin gradually wither and die, revealing the desert background. The withered plant remains and exposed soil forms a unique autumn landscape, rendering the contours of perennial plants like the Haloxylon ammodendron tree more distinct and in stark contrast with the soil background. In winter, the soil is covered by ice and snow, further accentuating the difference between Haloxylon ammodendron and the soil background, thereby reducing signal interference in remote sensing data. Consequently, data from the autumn and winter seasons are particularly important for predicting Haloxylon ammodendron tree biomass. This finding enriches the understanding of remote sensing applications in desert ecosystems.

Analysis of the sources of uncertainty

Conventional random cross-validation is appropriate for datasets exhibiting a relatively uniform distribution (De Bruin et al., 2022). When machine learning models are used in locations that are geographically distant from the initial training location, these models may face challenges, especially if the new region being predicted exhibits a distinct feature space. In the presence of spatial autocorrelation, the conventionally used random 10-fold CV may fail to provide an accurate assessment of model performance. In this study, the UAV sampling points were of an aggregated sampling type from a spatial distribution perspective, and they did not achieve uniform coverage of the study area. As a result, there were discrepancies in the feature space between the training instances and the forecasted targets. Such differences in feature space representation inevitably leads to model extrapolation, which can reduce the performance and transferability of regression modeling (Schmidt et al., 2014; Wadoux, Brus & Heuvelink, 2019). For strongly clustered sampling designs that predict most of the map through extrapolation, the use of block spatial cross-validation can ensure that the extrapolation results are closest to the accuracy metrics of the reference map. Limiting the prediction area can help avoid unreasonable extrapolation. This study employed the K-fold kNNDM spatial cross-validation method to establish a QRF quantile model. Through AOA analysis and dissimilarity index (DI) analysis, the prediction area was restricted to a range as similar to the training samples as possible, excluding areas with significantly different spatial features such as lakes, cultivated lands, wetlands, and salt lakes, thereby avoiding unreasonable model extrapolation.

Although the model’s relative error is high (SMAPE of 101%), combining the SMAPE and RAE from the final QRF model (spatial cross-validation) resulted in an absolute error smaller than the baseline model (RAE of 0.89; Fig. 8). Considering the low or zero values of biomass in the study area, any small prediction error would be amplified, resulting in a very high relative percentage error and a significant increase in SMAPE value. In this context, it may indicate that the model’s predictions are relatively accurate in regions with higher biomass values. To verify this hypothesis, the training data were divided into low-value and high-value categories based on the 50th percentile of biomass values. An analysis of the observed and predicted values for these two groups yielded the results in Fig. 10.

Figure 10 Comparision of metrics for low value region and high value region.

In the low-value region, although the absolute errors (MAE, RMSE) were smaller, the relative errors (RAE, SMAPE) were larger. This indicates that the model’s predictions in these regions were relatively accurate; however, due to the small actual values, even minor prediction errors appeared relatively large. In the high-value region, the absolute errors (MAE, RMSE) were larger, but the relative errors (RAE, SMAPE) were smaller. This indicates that, although the model’s prediction errors were larger in these regions, the error proportion was smaller relative to the actual values. The larger errors in the high-value region were usually accompanied by higher prediction uncertainty, as the model’s predictions in these areas exhibited not only larger errors but also higher volatility. This explains why the RMSE was higher in the high-value region and reflected greater prediction uncertainty.

Potential method improvements

By incorporating a blend of data from various satellites and on-site measurements, machine learning algorithms were used to create models for estimating biomass in regions characterized by extremely sparse vegetation. The findings suggest that the assessment of biomass in such arid regions using the methodology developed in this study presents potential opportunities despite notable uncertainties. Further endeavors are necessary to develop more resilient, pragmatic, and economical methodologies for estimating shrub AGB through satellite remote sensing. One potential avenue for further research involves the exploration of diverse machine learning models or the fusion of remote sensing physical models. Using ensemble machine learning models that combine predictions from various individual models may lead to more accurate forecasts (Hao et al., 2020; Robert et al., 2016). Additionally, incorporating the latest deep learning techniques could be beneficial as these techniques can handle complex, non-linear relationships in remote sensing data and reduce uncertainties in biomass assessment (Theron et al., 2022) . Models need to establish a balance between accuracy and generalization performance to achieve better outcomes (Schratz et al., 2019).

Selecting suitable remote sensing parameters is essential for the estimation of AGB (Lu et al., 2014). Different combinations of variables can influence how AGB relates to remote sensing data. These factors include the limitations of remote sensing data—such as spectral bands, spatial resolution, and availability—along with climatic, topographical, and soil conditions that are relevant to the plant growth environment (Zhao et al., 2016). Selecting appropriate variables is essential for enhancing the accuracy of predictions. Research indicates that simpler machine learning models, which are built with only a limited number of predictive variables, demonstrate better adaptability than intricate models (Lauria et al., 2015). The results of this study suggest that the training inputs should be limited to essential model features.

Ultimately, the model’s precision relies on the distribution and representative nature of the sampling sites. To enhance the model’s adaptability and effectiveness, as well as its applicability to surveys in different or larger regions, it is crucial to ensure a uniform distribution of training data across all areas. Examining specific regions where the model shows weaknesses could indicate potential areas for future sampling.

The impact and novelty of the findings

This study presents a novel mapping approach by integrating UAV LiDAR data with high-resolution, multi-temporal satellite images (Sentinel-1 and Sentinel-2) to estimate the aboveground biomass (AGB) of sparse shrub communities in arid regions. This combination capitalizes on the complementary strengths of different data sources. UAV LiDAR offers detailed structural information on a small scale, while satellite images provide large-scale coverage and multi-temporal perspectives. The application of a spatial cross-validation method and the quantile regression forest (QRF) prediction model further improved the accuracy and reliability of AGB estimates. This integrated methodology fills an important gap in previous research on desert shrub AGB estimation.

The results of this study offer insights into the ecological characteristics of the Haloxylon ammodendron forest in the Junggar Basin. Mapping the AGB distribution over 14,000 square kilometers revealed significant spatial heterogeneity, with biomass increasing from northwest to southeast. This significant spatial heterogeneity was closely related to environmental factors like elevation and soil conditions. Notably, the discovery of the crucial role of seasonal factors, especially autumn and winter data, in predicting AGB is novel,as it uncovers the unique ecological processes in the desert ecosystem, where the phenology of ephemeral plants interacts with the perennial Haloxylon ammodendron, influencing spectral characteristics and AGB estimation accuracy. This deepens the understanding of the complex dynamics and factors shaping vegetation growth and distribution in desert ecosystems.

Accurate estimation of Haloxylon ammodendron AGB and its spatial distribution is highly significant for ecosystem conservation in arid regions. The results of this study serve as a scientific foundation for formulating targeted conservation strategies. Areas with higher biomass can be designated as key protection zones for implementing measures like restricting human activities and enhancing ecological monitoring. In contrast, regions with lower biomass and higher uncertainty may require more intensive restoration efforts such as reforestation and soil improvement. This study also provides a quantitative tool for evaluating the effectiveness of conservation measures over time by monitoring AGB changes. Overall, this research contributes to the sustainable management and conservation of desert ecosystems, which are vulnerable to climate change and human impacts.

Conclusions

This study developed an innovative machine learning model integrating multi-source remote sensing data to predict the AGB distribution of Haloxylon ammodendron forest in the Manas Lake sedimentary plain, western Junggar Basin. This demonstrated the feasibility of satellite remote sensing for large-scale shrub AGB estimation in arid regions with low vegetation cover.

The results showed significant spatial heterogeneity of Haloxylon ammodendron forest biomass, increasing from northwest to southeast. The model, despite some prediction uncertainty, provides crucial data for understanding the spatial diversity of the ecosystem and evaluating the impacts of climate change and human activities on the local desert ecosystem, which could be used to guide ecological protection and management.

However, challenges remain in improving prediction accuracy. Future research should focus on optimizing methods to enhance model performance. This study contributes to the field by filling knowledge gaps and providing a basis for further studies on arid region desert ecosystems.

Additional Information and Declarations

Competing Interests

Author Contributions

Data Availability

The authors declare there are no competing interests.

XueFeng Yang conceived and designed the experiments, performed the experiments, analyzed the data, prepared figures and/or tables, authored or reviewed drafts of the article, and approved the final draft.

The following information was supplied regarding data availability:

The data is available at figshare: Xuefeng, Yang (2024). Vector data (ArcGIS Shp) files of Haloxylon ammodendron in the ancient lake basin area of Manas Lake in the western Junggar Basin. figshare. Dataset. https://doi.org/10.6084/m9.figshare.27890475.v1.

The data is also available at Zenodo: Yang, Xuefeng (2024). Map of Aboveground Biomass and Uncertainty of Haloxylon in the Ancient Manas Lake Basin Area, Western Junggar Basin, Xinjiang, China [Data set]. Zenodo. https://doi.org/10.5281/zenodo.12597120.

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
