# Peer review of "Mapping desert shrub aboveground biomass in the Junggar Basin, Xinjiang, China using quantile regression forest (QRF)"

_PeerJ, doi:10.7717/peerj.19099_

## Round 0.1 · original submission · Major Revisions

Add a section discussing the discoveries' significance and originality. Indicate the statistical techniques that were applied to the data analysis. Rewrite the findings to be clearer and more implications-focused.

Reviewer 1 ·

Basic reporting

1- The manuscript generally meets the standards for clear and professional English. However, there are instances of typographical errors and awkward phrasing that could be improved for clarity.

2- The literature references are adequate, providing sufficient background and context for the study.

Experimental design

The research presented is original and falls within the aims and scope of the journal. The research question is well-defined and relevant, addressing a significant knowledge gap regarding the aboveground biomass of Haloxylon ammodendron in arid regions.

Validity of the findings

Comment: The impact and novelty of the findings are not explicitly assessed in the manuscript. It would be beneficial to include a discussion on how the results contribute to the existing literature and the potential implications for ecosystem conservation strategies.

While the underlying data appear robust and statistically sound, the manuscript should explicitly state the statistical methods used to analyze the data. This would strengthen the validity of the findings.

The conclusions are generally well-stated and linked to the original research question. However, they could be more concise and focused on the implications of the findings rather than reiterating the results.

Suggested Improvements:

Include a discussion on the impact and novelty of the findings.
Clearly state the statistical methods used for data analysis.
Revise the conclusions to be more concise and focused on implications.

·

Basic reporting

See below

Experimental design

See below

Validity of the findings

See below

Additional comments

This paper takes the Haloxylon ammodendron forests in the Junggar Basin, Xinjiang, as the research subject, employing the Quantile Regression Forest (QRF) method to estimate aboveground biomass and performing a spatial consistency assessment. The authors' efforts are commendable. However, the paper presents the following issues:

1. Was all the work presented in the paper solely carried out by the author?


2. Title changed to: "Mapping Desert Shrub Aboveground Biomass in the Junggar Basin, Xinjiang using QRF."

3. Please add a title to Figure 1C.

4. Line 282 mentions that a total of 91 features were used to train the model, but not all of them are shown in Table 1. Where are the remaining features presented? Additionally, please reorganise Section 2.2.1 to clearly state how many optical indices were used and how many were derived from S1. Then, briefly explain the indices involved.

5. What is the title of each figure? I cannot see the precise descriptions of the figures on my end.

6. The spatial resolution of the ground AGB derived from remote sensing data is 600 metres. What is the spatial resolution of the ground AGB derived from LiDAR data? How do you ensure that the two datasets are compatible? LiDAR data typically has centimetre-level resolution, whereas the remote sensing data is mentioned to have a pixel size of 600 metres. How is the error between the two kept within a controllable range? This is crucial for the credibility of the final results. Please address this in the paper.

7. line 81, I think the sentences from here to the end of paragraph is quite related to the Sec. Research Area, please delete them.


8. In Section 2.3.6, please provide a detailed explanation of DI to help readers better understand the concept behind AOA.

9. Please make the corresponding changes to the x-axis titles of each plot in Figure 4; do not label them as 'value'.


10. In Section 3.5, please explain how random cross-validation was performed. Is this step for validating the trained model? The explanation is currently unclear and difficult to understand. If I understand you correctly, please divide this section into two subheadings: one for evaluating the trained model, where you can include a scatter plot with observed values on the x-axis and predicted values on the y-axis, along with the various metrics you mentioned in the text. The second subheading should focus on spatial validation, where AOA and DI are calculated.

11. For Figures 9a and 9b, please replace the legend labels from 'low' and 'high' with the specific values. Since the entire paper emphasizes the calculation of shrub AGB, it would be more appropriate to provide the actual values.

12. Line 386, "figure 6 seasons?" Where is this mentioned earlier in the text? It is quite confusing. Are they one of the training features?

---

## Round 0.2 · accepted · Accept

This revised version is suitable for publication in PeerJ.

·

Basic reporting

The author gave a good response based on my last question. I suggest accepting and publishing it.

Experimental design

The experimental design is sophisticated and rigorous, taking into full consideration the principles of variable control and randomization, ensuring the robustness and repeatability of the research results.

Validity of the findings

The experimental data showed good consistency and repeatability, indicating that the research conclusions have wide applicability and promotion value